# GEM: Gestalt Enhanced Markup Language Model for Web Understanding via Render Tree

**Zirui Shao**[1*], **Feiyu Gao**[2*], **Zhongda Qi**[1], **Hangdi Xing**[1],
**Jiajun Bu**[1], **Zhi Yu**[1†], **Qi Zheng**[3], **Xiaozhong Liu**[3]

[1]Zhejiang Provincial Key Laboratory of Service Robot, Zhejiang University
[2]Alibaba Group [3]Worcester Polytechnic Institute
{shaozirui, qzd, xinghd, bjj, yuzhirenzhe}@zju.edu.cn
feiyu.gfy@alibaba-inc.com, yongqi.zq@taobao.com, xliu14@wpi.edu

## Abstract

Inexhaustible web content carries abundant perceptible information beyond text. Unfortunately, most prior efforts in pre-trained Language Models (LMs) ignore such cyberrichness, while few of them only employ plain HTMLs, and crucial information in the rendered web, such as visual, layout, and style, are excluded. Intuitively, those perceptible web information can provide essential intelligence to facilitate content understanding tasks. This study presents an innovative Gestalt Enhanced Markup (GEM) Language Model inspired by Gestalt psychological theory for hosting heterogeneous visual information from the render tree into the language model without requiring additional visual input. Comprehensive experiments on multiple downstream tasks, i.e., web question answering and web information extraction, validate GEM superiority.

## 1 Introduction

Web pages serve as crucial carriers for humans to acquire and perceive information. Due to the significant wealth of these documents, long-standing efforts have been undertaken to address web understanding tasks (Chen et al., 2021; Hao et al., 2011; Dong et al., 2014; Escudeiro and Escudeiro, 2009; SnehaY. et al., 2012). However, understanding web pages can be challenging for automated systems compared to humans, as the design of layout and visual style caters specifically to human perceptual patterns, thereby facilitating comprehension.

Recently, the pre-trained Language Models (LMs) (Li et al., 2022; Deng et al., 2022) advance web understanding and demonstrate superior performance on various related tasks by jointly pre-training on text and markup information. Nevertheless, these models oversimplify web pages as plain HTML[1] (HyperText Markup Language) sequences

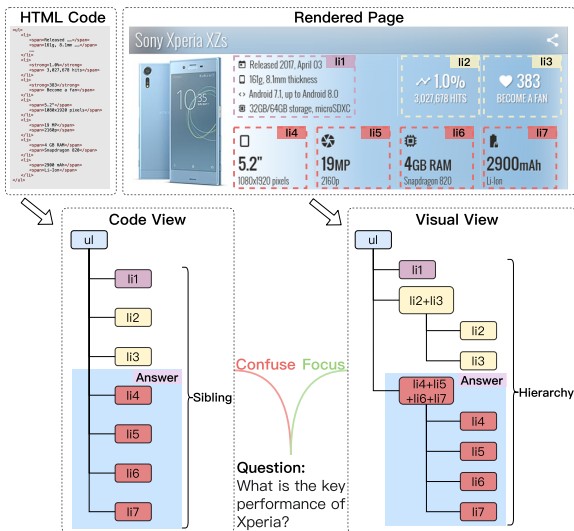

Figure 1: From the HTML code view, the page consists of seven siblings <li>, which can be divided into three groups from the visual perception view (represented by differently colored boxes). By taking hierarchical relationships of the visual view into account, the system can answer the question "*What is the key performance of the Xperia?*" better than only considering the sibling relationship.

while neglecting the advantageous visual information of rendered pages, such as style and layout, which is essential for web understanding. Several recent works investigate web pages as pictures of screenshots (Xu et al., 2020b,a; Vishwanath et al., 2018; Garncarek et al., 2020; Huang et al., 2022), emphasizing the importance of the fixed layout of the input image. However, the dynamic nature of web page rendering can lead to significant appearance variations across devices and browsers, resulting in the limited applicability of such models. Additionally, these methods require an OCR[2] system to extract text with coordinates, which leads to extra overhead and neglect the hierarchical information provided by the markup language.

---

[*]Equal contribution.
[†]Corresponding author.
[1]https://en.wikipedia.org/wiki/HTML

[2]https://en.wikipedia.org/wiki/Optical_character_recognition

Regardless of render conditions, humans are capable of rapidly comprehending the content of rendered web pages, which indicates that human visual perception is an effective way to consume web semantics (Xiang et al., 2007; Xu and Miller, 2016). To leverage this advancement, we propose to encode visual information of web pages into the LM flexibly via render-tree-enabled pre-training tasks. It is inspired by human perceptive patterns rather than taking visual features as input directly. Gestalt psychological theory (Wertheimer, 1938; Koffka, 1955), a prominent cognitive model, explains the human perceptive processes that elements with similar visual styles and proximate locations are commonly regarded to be of similar semantic functions. For instance, as Figure 1 depicts, while the seven nodes of a given web page are siblings in the markup language, their different rendering appearances and positions categorize them into three semantic blocks. Without visual awareness (i.e., only using HTML tags), it is challenging to answer downstream questions like "*What is the key performance of Xperia?*". However, when considering the visual cues of the web page, particularly the answer is comprised of nodes in the same row and color, this question becomes rather achievable.

In this paper, we propose **G**estalt **E**nhanced **M**arkup (GEM) language model designed to host visual knowledge into the language model without additional visual input requirement. The GEM utilizes visual information of rendered web pages solely during the pre-training, while during fine-tuning or inference, the rendering is unnecessary.

To incorporate visual perception into language models, two further pre-training objectives are proposed, each relating to one Gestalt law. The one is Same Textual Style Prediction (STSP), which is based on the Gestalt laws of similarity that humans tend to perceive objects with similar appearances as a group. This task enlightens the language model with an appearance perception to learn semantics. On the other hand, the Proximate Nodes Prediction (PNP) enables the model to understand the relationships of elements from visual positions, not only from the DOM[3] (Document Object Model) tree. The PNP task is proposed on the Gestalt proximity law, according to which humans consider objects close to each other perceptively coherent. Our corpora are built from render trees of rendered web pages, which combine the DOM and CSSOM[4] (CSS Object Model) that contain comprehensive information about rendered web pages, including textual, structural, and visual information. In practice, GEM can learn a strengthened representation of markup language with the enhancement of visual prior knowledge.

To validate GEM superiority, we conduct experiments on two downstream tasks, i.e., *web question answering* and *web information extraction*. GEM consistently surpasses several strong baselines. Moreover, we verify the model architecture adaptability of Gestalt objectives. In addition, we compare the performance of GEM and large language models (LLMs) (Brown et al., 2020; Ouyang et al., 2022a; Chowdhery et al., 2022).

Our main contributions are as follows:

- The proposed GEM model introduces a render tree as a powerful approach to enhance the pre-training of language models with considerable visual knowledge acquired from web pages.

- Based on Gestalt psychological theory, two innovative Gestalt pre-training objectives have been proposed to enable visual perception of GEM, which has been proven beneficial for various downstream tasks.

- The pre-trained model and code of GEM are publicly available at GitHub[5].

## 2 Preliminaries

### 2.1 Review of Gestalt Theory

Gestalt, a German word, is referred to "unified whole". The Gestalt laws proposed by German psychologist Max Wertheimer (Wertheimer, 1938) describe how humans group elements in perception. Graphic designers use these laws to arrange elements on web pages and other interfaces (Graham, 2008). Web pages that violate the Gestalt laws result in comprehension difficulties due to mismatched semantic and perceived structures (Sani and Shokooh, 2016). Thus, such pages are likely to be phased out (Xiang et al., 2007). We can assume modern pages are mostly well-designed and follow Gestalt laws.

---

[3]https://en.wikipedia.org/wiki/Document_Object_t_Model

[4]https://www.w3.org/TR/cssom-1/

[5]https://github.com/AlibabaResearch/AdvancedLiterateMachinery/tree/main/DocumentUnderstanding/GEM

This paper applies two Gestalt laws, similarity and proximity, to enhance the LM with visual perception, as detailed below.

- **The Gestalt law of similarity.** According to this law, similar objects are perceptually grouped (Wertheimer, 1938). In web pages, the similarity is based on rendered appearance. For instance, in Figure 2, the font color of the upper two yellow-framed nodes is white, forming a group related to the popularity of the phone.

- **The Gestalt law of proximity.** This law states that nearby objects are perceptually grouped (Wertheimer, 1938). In web pages, proximity is measured by rendered positions. For instance, in Figure 2, the lower two green-framed nodes are close, creating a group that highlights the key performance of the phone.

Based on this knowledge, pre-training tasks can be developed according to the Gestalt laws, enabling the language model to simulate the perceptive processes of humans and better understand the semantic relationships among web contents.

## 2.2 Render Tree

The render tree[6] is of great significance in rendering web pages, which is processed through four steps. The first two steps parse HTML and CSS[7] (Cascading Style Sheets) documents to create DOM and CSSOM trees, which are independent objects describing content and style rules. In the third step, DOM and CSSOM are merged into the render tree by reserving all the visible DOM nodes and mounting CSSOM style information to the corresponding node. Eventually, the browser traverses the render tree, calculates each node's exact size and position, and transforms nodes to actual pixels on the screen. In brief, the concept of render trees contains comprehensive information on rendered web pages, encompassing textual content, HTML structure, stylistic, positional, and other visual information. In this study, render trees are employed to build the corpora for pre-training.

---

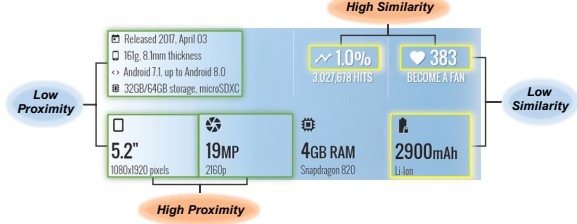

Figure 2: Illustrations of the Gestalt law of similarity and proximity.

## 3 Methodology

### 3.1 Preparing Inputs

Web pages are easily gathered at scale, yet they can not provide straightforward supervision for visual perception. Thus, several pre-processings are conducted during pre-training. We filter out pages that are non-renderable using *headless chrome*[8] and render the remaining ones. Next, we randomly sample a rendered page from the remaining pool, which can avoid any bias towards certain websites or domains and ensure the diversity of the data. Additionally, *selenium*[9] is applied to store HTML containing only visible nodes and to record each node's specified CSS properties (required for pre-training). Since many web pages retain thousands of tokens, we truncate them into sub-pages employing sliding windows proposed by Deng et al. (2022) . As shown in Figure 3, GEM takes HTML as input, which is processed into text tokens and corresponding XPath[10] (XML Path Language) expressions. The CSS properties are utilized as the ground truth for the Gestalt pre-training tasks that are covered in section 3.2.

### 3.2 Pre-training Objectives

As Figure 3 depicts, we propose two Gestalt objectives to inject visual awareness into the language model. To preserve contextual and hierarchical information, GEM also employs the markup objectives proposed by Li et al. (2022). The ultimate pre-training objective is the summation of the markup objectives and the Gestalt objectives.

The following two subsections elaborate on the proposed Gestalt objectives respectively.

---

[6] https://web.dev/critical-rendering-path-render-tree-construction/

[7] https://en.wikipedia.org/wiki/CSS

[8] https://developer.chrome.com/blog/headless-chrome/

[9] https://www.selenium.dev/

[10] https://en.wikipedia.org/wiki/XPath

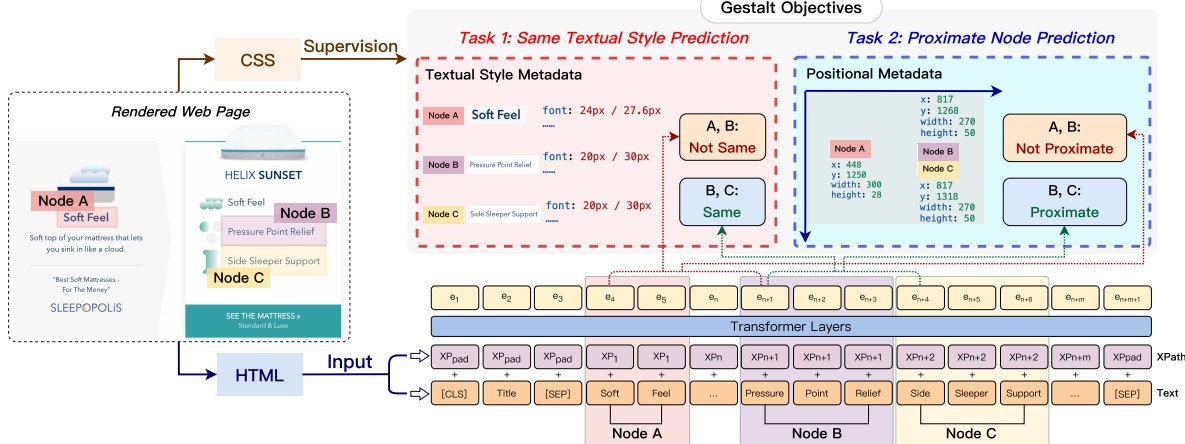

Figure 3: An overview of the proposed GEM. Given a rendered web page, a browser is utilized to obtain its render tree and automatically extract HTML and CSS properties. The HTML is utilized as input, which is further processed into text tokens and XPath expressions, while CSS properties are employed as supervision for two Gestalt objectives. These objectives include Same Textual Style Prediction (STSP), based on the law of similarity, and Proximity Node Prediction (PNP), based on the law of proximity. STSP equips the LM with visual-appearance perception by predicting whether the textual styles of sampled node pairs are the same, while PNP facilitates the LM to perceive visual-position relationships by learning whether sampled node pairs are proximate. Note that the markup objectives of GEM are hidden in this figure.

### 3.2.1 Same Textual Style Prediction

The first Gestalt objective is Same Textual Style Prediction (STSP), utilizing the Gestalt law of similarity (see Section 2.1). The similarity of web elements is defined by their appearances, which consist of size, background, and foreground (Xu and Miller, 2016). For the sake of simplicity, hereafter, those attributes are collectively called **textual style**. Referring to CSS Reference[11], we figure CSS properties of font, color, and background-color as joint control of textual style.

In pre-training, the Gestalt law of similarity is translated into classifying a pair of text nodes with "same" or "not same" textual style. A given pair of text nodes are considered the same style if all textual-style-controlling CSS properties are the same. For instance, in Figure 3, Node $A$ and Node $B$ are assigned the "not same" label due to their non-identical font size. We randomly sample node pairs from one page, and the model is required to classify the pairs with the features from the first token of each node.

The STSP task provides supervision for textual-style-similarity clues, which enables GEM to understand semantic relationships by incorporating prior knowledge of textual-style design.

### 3.2.2 Proximate Nodes Prediction

Besides the textual style information, the visual-position relationships of nodes are also essential for visual awareness in web understanding. Hence, the Proximate Node Prediction (PNP) objective is proposed, leveraging the Gestalt law of proximity (see Section 2.1). In web pages, the rendering regions of nodes are extracted from the render tree to evaluate their proximity. Notably, these rendering regions incorporate the padding, which is the space between the rendering border and content.

In pre-training, the proximity is interpreted by comparing the position of nodes' edges (i.e., left, top, right, and bottom) that are calculated based on their rendering regions. If any edges of two nodes share the same value, they are considered proximate. As an example in Figure 3, the bottom edge of Node $B$ and the top edge of Node $C$ coincide, indicating they are proximate. Conversely, Node $A$ and Node $B$ share no edges and thus are non-proximate. We randomly sample node pairs and ask the model to predict if they are proximate using the features of the first token from each node.

Take a situation where nodes are far in the DOM tree but visually close, providing a vital clue that these nodes may be semantically related. The PNP task enables GEM to consider the semantic relationships among nodes utilizing the prior knowledge of proximity design apart from the structure of the

---

[11] https://www.w3schools.com/cssref/index.php

DOM tree.

## 4 Experiment

### 4.1 Pre-training Setups

#### 4.1.1 Data

Our corpora are built from the Common Crawl[12] dataset. We derive approximately 2 million training samples from 100k renderable web pages by pre-processing. Details of the data pre-processing are available in Section 3.1. We recognize the possibility of noisy samples in our corpus that don't adhere to the Gestalt principles. However, such instances are minimal since, as demonstrated by Xiang et al. (2007), web pages that contradict the Gestalt principles typically experience short lifespans on the internet. In order to ensure fairness, we remove all the web pages that appear in the downstream task datasets. The settings of markup objectives follow Li et al. (2022). In both the STSP and PNP tasks, we initially traverse and label all node pairs in a given training sample as positive or negative, and store them in separate pools. Subsequently, we randomly sample 100 samples from each pool (totaling 200 samples) for pre-training. If either pool has fewer than 100 samples, we employ oversampling to guarantee data balance. Using this approach, each node pair has an equal probability of being labeled as "same" in STSP and as "proximate" in PNP.

#### 4.1.2 Implementation

We set up GEM following the MarkupLM$_{base}$ and initialize it with the pre-trained weight provided by Li et al. (2022). Additionally, we also implement a RoBERTa$_{base}$-based GEM named "GEM-R" on the same corpora. MarkupLM enhances RoBERTa by incorporating a new XPath embedding layer to model HTML structures. GEM-R is pre-trained using the MLM objective and the Gestalt objectives. As in the GEM, we utilize the first token features as the node features in GEM-R. The input format of GEM-R is the same as T-PLM (Chen et al., 2021), i.e., pure text extracted from HTML. Our implementation uses Adam optimizer (Kingma and Ba, 2014) with a learning rate of 1e-5 and a batch size of 128 training samples with a maximum of 384 tokens. The pre-training is done on 8 Nvidia-V100 GPUs for 300K steps. We evaluate the pre-training performance in the Appendix A.5.

---

[12] https://commoncrawl.org/

### 4.2 Fine-tuning

We experiment on two downstream tasks to evaluate GEM: *web question answering* and *web information extraction*. Note that both GEM and GEM-R do not require rendered web pages in fine-tuning.

#### 4.2.1 Web Question Answering

Web question answering is a task that automatically answers questions about a given web page, which requires a system to comprehensively understand the spatial and logical structure of the web page. We employ the Web-based Structural Reading Comprehension (WebSRC) dataset (Chen et al., 2021) to verify the ability of GEM. WebSRC contains 400K question-answer pairs from 6.5K web pages and provides corresponding HTML source codes, screenshots, and metadata. The answers are either text spans on pages or yes/no. We follow previous work (Li et al., 2022) to take WebSRC as a typical extractive reading comprehension task, in which the token representations are fed into an output layer to predict the start and end indexes of the answer (Devlin et al., 2019). The evaluation metrics are **Exact match (EM)**, **F1 score (F1)**, and **Path overlap score (POS)**, as defined in the original paper (Chen et al., 2021), where **POS** is a tag level metric that measures the accuracy of locating HTML tags. We follow Li et al. (2022) to conduct the experiment on the official train/dev sets and report the results on the development set. We fine-tune the pre-trained LMs for 10 epochs with a batch size of 32, and the learning rate is 1e-5.

#### 4.2.2 Web Information Extraction

We use the Structure Web Data Extraction (SWDE) dataset (Hao et al., 2011) to evaluate GEM, a real-world web page collection for automatic information extraction on the web. The SWDE consists of over 124k web pages from 80 websites of 8 verticals (10 websites per vertical). The task requires the model to extract the values for several given attributes (3 to 5 different attributes according to the vertical) from a page. In the actual application scenario, due to labor costs, only limited labeled data can be obtained for a given vertical. However, the system is required to work on a much larger website set. Thus, we evaluate GEM on each vertical independently with a few-shot setting, where 50% of websites are training data, and the rest are testing data. Note that websites in the testing set are unseen during training. **Precision**, **Recall**, and

| Category | Model | EM | F1 | POS |
|---|---|---|---|---|
| Pure Text | T-PLM | 52.12 | 61.57 | 79.74 |
| | RoBERTa$_{base}$ | 51.89 | 62.48 | 80.11 |
| | RoBERTa$_{base}$* | 52.56 | 62.61 | 80.38 |
| | GEM-R | 54.88 | 64.81 | 81.60 |
| HTML (H) | H-PLM | 61.51 | 67.04 | 82.97 |
| | MarkupLM$_{base}$ | 66.67 | 73.74 | 87.47 |
| | MarkupLM$_{base}$* | 66.83 | 73.24 | 86.63 |
| | GEM | **69.12** | **75.93** | **88.41** |
| H+Visual | V-PLM | 62.07 | 66.66 | 83.64 |

Table 1: Web question answering results on the Web-SRC dataset, categorized by input modality. "*" means further pre-training on our corpora with the original objective only. Underlined figures represent the best results of the text input group, while bold figures indicate the best results of all models.

**F1 score** on page level are evaluation metrics on this task, following Hao et al. (2011). To make a fair comparison, we follow Zhou et al. (2021) to pre- and post-process data. The results of each vertical are the average of 10 training set permutations. The final experiment results are obtained by taking the average of all 8 verticals. We fine-tune the pre-trained LMs for 10 epochs with a batch size of 64, and the learning rate is 2e-5.

### 4.3 Results

#### 4.3.1 Web Question Answering

The results of web question answering are shown in Table 1. Several baselines are compared, which are categorized into three groups based on their input modality: (1) *Pure Text*: T-PLM, RoBERTa, and GEM-R utilize non-structural pure text by deleting all HTML tags follow Chen et al. (2021); (2) *HTML*: H-PLM, MarkupLM, and GEM take HTML as input, using different per-processing methods, where H-PLM follows Chen et al. (2021) and MarkupLM, GEM follow Li et al. (2022). (3) *HTML+Visual*: V-PLM (Chen et al., 2021) leverages both HTML and screenshots as inputs. Since GEM does not have visual input, other strong models containing visual features or metadata attained by rendered web pages (e.g., coordinates of elements) are not included.

As depicted in the results, GEM consistently surpasses other baselines. The gap between GEM and MarkupLM* illustrates the effectiveness of incorporating visual awareness through Gestalt tasks.

| Category | Model | P | R | F1 |
|---|---|---|---|---|
| Non-PLM-based | SSM | - | - | 74.10 |
| | FreeDOM-Full | - | - | 92.56 |
| | SimpDOM | - | - | 93.75 |
| PLMs with pure text | RoBERTa$_{base}$ | 95.31 (±1.26) | 93.55 (±1.99) | 94.05 (±1.80) |
| | RoBERTa$_{base}$* | 95.43 (±1.62) | 93.46 (±2.34) | 94.09 (±2.15) |
| | GEM-R | 95.91 (±1.25) | 94.05 (±1.77) | 94.57 (±1.60) |
| PLMs with HTML | MarkupLM$_{base}$ | 95.99 (±1.49) | 95.16 (±1.70) | 95.57 (±1.65) |
| | MarkupLM$_{base}$* | 96.04 (±1.54) | 95.14 (±1.87) | 95.59 (±1.81) |
| | GEM | **96.84 (±1.42)** | **95.66 (±1.59)** | **96.04 (±1.53)** |

Table 2: Evaluation results on web information extraction task (SWDE dataset), categorized by method type. Some P (Precision) and R (Recall) values are left blank due to unreported in original papers. Each value is reported as "mean ± standard deviation" calculated from 80 experiments. "*" stands for the model further pre-trained on our corpora with its original objective only. Underlined figures represent the best results of the text input group, while bold figures indicate the best results of all models.

Moreover, GEM-R achieves the best results among all models with text input, indicating the robust adaptability of the Gestalt tasks in model architecture. Notably, GEM-R does not incorporate an XPath embedding layer and relies solely on non-structural pure text as input, suggesting that Gestalt tasks can capture visual information without the explicit incorporation of HTML structure. Additionally, unlike V-PLM, which relies on input screenshots for visual perception, GEM solely utilizes HTML as input and does not require rendered web pages during fine-tuning and inference, which provides significant deployment convenience advantages while preserving visual perception.

#### 4.3.2 Web Information Extraction

The results of web information extraction are in Table 2, where the compared models are classified into three groups: (1) *Non-PLM-based methods*: SSM (Carlson and Schafer, 2008), FreeDOM-Full (Lin et al., 2020), SimpDOM (Zhou et al., 2021); (2) *PLMs with pure text as input*: RoBERTa, GEM-R are detailed in Section 4.3.1; (3) *PLMs with*

| WebSRC Challenge Set | | | |
|---|---|---|---|
| Model | EM | F1 | POS |
| GEM | 67.04 | 72.69 | 87.00 |
| GPT-3.5-turbo | 66.99 | 71.92 | 68.43 |
| Llama2 | 25.91 | 31.54 | 37.97 |
| Llama2-FT | 48.95 | 53.50 | 65.17 |
| SWDE Random Subsets | | | |
| | Precision | Recall | F1 |
| GEM | 97.06 | 96.09 | 96.36 |
| GPT-3.5-turbo | 28.61 | 25.93 | 26.18 |
| GPT-3.5-turbo* | 27.54 | 26.61 | 26.35 |
| Llama2 | 25.98 | 26.10 | 26.04 |
| Llama2* | 26.27 | 26.10 | 26.06 |

Table 3: Performance comparison of LLMs and GEM on downstream tasks, with retested results for GEM on WebSRC Challenge Set and SWDE Random Set. "Llama2-FT" denotes Llama2 fine-tuned on the Web-SRC training set. The prompt of "*" is limited to the text of nodes.

*HTML as input*: MarkupLM and GEM, which are the same as Section 4.3.1.

GEM achieves similar results on web information extraction as on web question answering. GEM outperforms all baselines, and GEM-R is the best model in the "*PLMs with pure text*" group. The improvement achieved by GEM is significant, considering that the baselines demonstrate remarkable performance, and each value in Table 2 is the average of 80 experiments. Another notable observation is that Gestalt tasks decrease the standard deviation of the results, indicating that incorporating visual perception can enhance the robustness of models.

## 4.4 Discussion on Large Language Model (LLM)

Recently, large language models (LLMs) have been gaining adoption in different domains. Hence, we assess LLM on both downstream tasks, as shown in Table 3. We compare two baselines: GPT-3.5-turbo (Ouyang et al., 2022b; Brown et al., 2020) and Llama2 (Touvron et al., 2023). GPT-3.5-turbo represents one of the current state-of-the-art LLMs and is accessible via the OpenAI API[13]. On the other hand, Llama2 is a prevalent open-source large model in academia.

The specific pre-trained model weight we utilize, *Llama-2-7b-chat-hf*, is available on Huggingface[14].

The objective of the Web Question Answering task (with the WebSRC dataset) is to explore the model's capacity to comprehend the spatial and logical structure of a given web page. LLM, hosting an extensive knowledge repository, can answer general questions (Petroni et al., 2019), regardless of the given web page. To fit the project scope, we employ a subset from WebSRC, tagged as the "*challenge set*", by eliminating questions that GPT-3.5 can respond to correctly based solely on the question content. The challenge set comprises over 5,000 questions, with further details provided in Appendix A.1.

The prompt template for WebSRC employs in-context learning (Brown et al., 2020), incorporating a task description, selected demonstrations, and a test instance. For each question-page pair, we randomly select $n$ demonstrations from the same vertical to ensure semantic relevance. Due to the token limit, we set $n$ to 3. Further details are available in Appendix A.2. Additionally, to address discrepancies in training data, we fine-tune Llama2 using the WebSRC training set. Further details about Llama2 fine-tuning are in the Appendix A.3.

As shown in Table 3, GEM and GPT-3.5 achieve comparable performance on Exact Match and F1 score, on POS, GEM significantly outperforms GPT-3.5. Regarding Llama2, whether or not it undergoes fine-tuning, GEM significantly surpasses it. The fine-tuning outcomes align with expectations that fine-tuning improves Llama2's performance on the WebSRC dataset. The experimental results highlight GEM's superiority in consuming HTML structural information and validate LLM's incapability to provide answers despite access to the highly pertinent web page, mainly attributed to the LLM's lack of comprehension and familiarity with the HTML structure.

For the web information extraction experiment using the SWDE dataset, considering that the SWDE requires 80 experiments and its web pages are of considerable length, we, due to limited resources, randomly select 800 samples for testing and do not conduct experiments on fine-tuning Llama2. The prompt template adheres to the in-context learning principle, with 3 randomly selected demonstrates. All web pages in the prompt

---

[13]https://platform.openai.com/

[14]https://huggingface.co/meta-llama/Llama-2-7b-chat

| | Objectives | | | Dataset | | | | | |
| | | | | WebSRC | | | SWDE | | |
| # | Markup | STSP | PNP | EM | F1 | POS | P | R | F1 |
|---|---|---|---|---|---|---|---|---|---|
| 1 | ✓ | | | 66.83 | 73.24 | 86.63 | 96.04 (±1.54) | 95.14 (±1.87) | 95.59 (±1.81) |
| 2 | ✓ | ✓ | | 67.78 | 74.27 | 87.55 | 96.58 (±1.10) | 95.26 (±1.49) | 95.64 (±1.39) |
| 3 | ✓ | | ✓ | 67.58 | 74.15 | 86.68 | 96.62 (±1.20) | 95.48 (±1.73) | 95.81 (±1.64) |
| 4 | ✓ | ✓ | ✓ | 69.12 | 75.93 | 88.41 | 96.84 (±1.42) | 95.66 (±1.59) | 96.04 (±1.53) |

Table 4: Ablation study on the WebSRC and SWDE dataset. "STSP" and "PNP" stand for Same Textual Style Prediction and Proximate Nodes Prediction. "Markup" denotes the original pre-training objectives of MarkupLM. The results on the SWDE dataset are reported as "mean ± standard deviation" over 80 experiments.

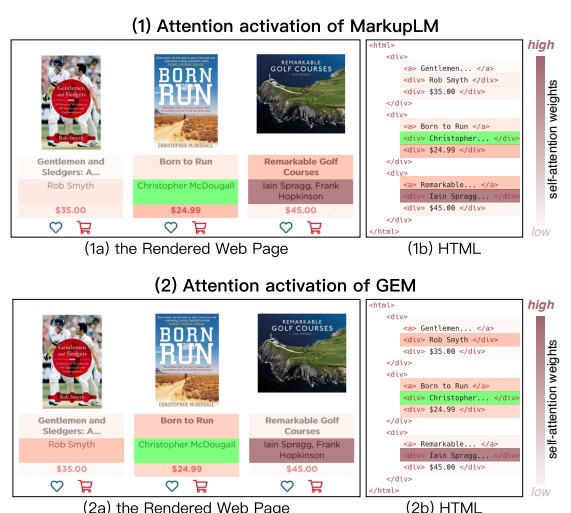

Figure 4: Visualization of node-level self-attention weights between the green node and other nodes, shown in both the rendered page and HTML (model's input).

undergo pre-processing according to Zhou et al. (2021), producing XPath-text node pairs. Note that the pre-processed web content consists of a set of segments rather than the entire web page, which is consistent with the input of the baselines described in Section 4.3.2. Moreover, to eliminate XPath's potential noise, we conduct an experiment with the restriction that only the text of nodes is provided. The prompt template details are in Appendix A.4. Experiment results confirm LLM's incapability for HTML structure comprehension.

## 4.5 Ablation Study

To further investigate the effectiveness of GEM, we perform a series of ablation experiments, as shown in Table 4. The models with different objectives are pre-trained in the same settings as in Section 4.1.2. Experiment results validate our hypothetical proposition: both style and positional information extracted from rendered web pages benefit web understanding. For Instance, in terms of the EM metric on the WebSRC, the proposed STSP shows an improvement of 0.95%, while the improvement achieved with the PNP is 0.75%. Meanwhile, the resonance between positional (PNP) and style (STSP) tasks and multi-task co-training can be vital to enhance the model performance. The models trained by an individual Gestalt task, without investigating multi-view landscapes, can hardly achieve superiority compared to the #4 model with holistic perception.

## 4.6 Attention Maps Visualization

To further investigate the effect of injected visual awareness, we choose a case in WebSRC as an example and visualize the attention activation between the green node and other nodes in the last layer of the encoder. The checkpoints we used are #1 and #4 in Table 4, which have not been fine-tuned. Both MarkupLM and GEM utilize HTML as their input, and to facilitate understanding, we provide visualizations on the rendered pages, as shown in Figure 4. The color blocks are manually painted using the activation of the first token of each node, which serves as the node representation in pre-training. MarkupLM essentially relies on the DOM tree since the high activation nodes are sequential in HTML code (shown in Figure 4 1b). In contrast, GEM incorporates additional vi-

sual prior knowledge beyond the DOM tree. Even without visual input, GEM pays more attention to nodes that are close to the green node, as well as those with similar textual styles (as displayed in Figure 4 2a). Visualization results demonstrate that the Gestalt tasks modify the attention mechanisms as intended. GEM's attention map aligns with the web page's visual appearance, indicating that GEM can establish connections between visual and structural/semantic information of the DOM tree. The attention mechanism of GEM is beneficial for web understanding, as verified in Section 4.3.1.

## 5 Related Work

Web pages contain rich perceptible information beyond text. Extracting this information is crucial for web understanding but also challenging. Previous works on this task mainly use rule-based modules (Soderland, 1999; Cohen et al., 2002; Gulhane et al., 2011; Hao et al., 2019), which are not robust to different websites.

Representation learning for plain text documents has been well-studied. Pre-trained Language Models (PLMs), which use text encoders with self-supervised objectives, achieve remarkable performance on numerous NLP tasks (Devlin et al., 2019; Liu et al., 2019; Lewis et al., 2019). Some recent works extend these methods to web pages by taking HTML documents as input and encoding semantic information with specific pre-training tasks (Li et al., 2022; Deng et al., 2022).

However, these models ignore the visual features of web pages, which are essential for understanding them. Some other works address this issue by treating web pages as images of screenshots (Xu et al., 2020b,a), but they lost the hierarchical structure of HTML. Zhao et al. (2022) and Xie et al. (2021) incorporate coordinates of web elements into the input along with the HTML. However, they all rely on the fixed layout of the input image, which could vary with different devices and browsers. Therefore, these models have limited applicability.

Our work diverges from prior attempts in two crucial areas: First, we model web pages by leveraging both HTML documents and render trees, providing the LM with visual perception. Additionally, we harness knowledge from web page visual aspects by encoding it into the text representation rather than using visual information as input.

Recent developments in LLMs (Brown et al., 2020; Ouyang et al., 2022a; Chowdhery et al., 2022) ushered in a new age of AI applications. While these advances have demonstrated a remarkable capacity, there remains an acute need for a comprehensive evaluation of their application to web understanding.

## 6 Conclusion

This paper presents GEM, a Gestalt Enhanced Markup (GEM) Language Model that leverages Gestalt psychological theory. GEM innovatively enriches the language model with heterogeneous visual information from render trees of web pages without requiring visual modality input. As part of this innovation, two distinctive Gestalt pre-training objectives are formulated in order to codify visual qualities such as style and position within the language model. Evaluations of different downstream tasks and backbones show that GEM can learn a stronger representation of markup language with visual knowledge enhancement.

## Limitations

Despite the success of the model GEM, some drawbacks need to be addressed in the future. Primarily, the proposed model relies on the render trees of web pages, which may not be immediately available or perfectly precise for certain complex or advanced web pages. Additionally, the potential of large language models (LLMs) is undeniable, and there is much incentive to exploit it. This paper refrains from integrating GEM with LLMs, as it represents a challenging prospect that requires extensive research. We aim to delve more deeply into this problem and make pioneering progress in our future research.

## Acknowledgements

This work is supported by the National Natural Science Foundation of China (Grant No. 61972349 ), the National Key R&D Program of China (No. 2018YFC2002603), and Alibaba-Zhejiang University Joint Institute of Frontier Technologies.

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

# A    Appendix

## A.1    WebSRC Challenge Set Selection

Due to limited cost, we randomly select 18,131 questions from the WebSRC development set, which contains 52,826 samples. We utilize the prompt defined in Table 5 to query GPT-3.5 and then compared the responses to the ground truth by calculating their similarity using the formula:

$$similarity = \frac{l_r + l_g - d}{l_r + l_g}$$

where $l_r$ and $l_g$ are the lengths of the responses and ground truth, whereas $d$ denotes the Levenshtein Distance (Levenshtein, 1966) between them.

Ultimately, we select 5,324 questions with a similarity below 20% for the challenge set.

## A.2    The Prompt Template of WebSRC

To evaluate LLM on WebSRC, we define the prompt template as shown in Table 7.

## A.3    Details of Fine-tuning Llama2 on WebSRC

We fine-tune Llama2 model utilizing Parameter-Efficient Fine-Tuning (PEFT) and Low-Rank Adaptation (LoRa) (Hu et al., 2021) techniques with the entire training set of WebSRC. The fine-tuning is done on 2 NVIDIA A100 GPUs for 1 epoch.

## A.4    The Prompt Template of SWDE

To evaluate LLM on SWDE, we define the prompt template as shown in Table 8. In order to facilitate understanding, a web page is sampled from the "auto" vertical, and the values of all slots in the sampled web page are presented in Table 9.

## A.5    Pre-training Performance Evaluation

We evaluate the pre-training performance of the two innovative pre-training tasks, STSP and PNP, both of which are binary classification tasks. For

| Prompt | Here is a question from a web page titled {Title} Question: {Question} You need to answer the question. The format of your reply: answer:[ANSWER]. And do not reply to other content. | |
|---|---|---|
| Slots | Title | The title of the given web page from WebSRC. |
| | Question | The question about the given web page from WebSRC. |

Table 5: The prompt for WebSRC Challenge Set selection.

| Task | P | R | F1 |
|---|---|---|---|
| STSP | 61.70 | 76.24 | 65.15 |
| PNP | 72.33 | 65.59 | 84.60 |

Table 6: Pre-training performance evaluation on held-out training data.

a comprehensive evaluation, we reserve a subset of the training data as the evaluation set. We use classification metrics, and the results are presented in the Table 6.

| Prompt | Here is a question and its corresponding page. |
|---|---|
| | You need to answer the question, and the ANSWER are either text spans on page or yes/no. You need to answer the TID, which is the deepest tag in the DOM tree which contain all the answer. For yes/no question, there is no tag associated with the answer, so the TID is -1.You also need to answer the STARTING_INDEX of the answer, which is the char offset of the answer from the start of the content of the tag specified by TID. Note that before counting this number, we first eliminate all the inner tags in the specified tag and replace all the consecutive whitespaces with one space. For yes/no questions, STARTING_INDEX is 1 for answer "yes" and 0 for answer "no". |
| | The format of your reply: answer: [ANSWER] tid: [TID] starting_index: [START-ING_INDEX]. And do not reply other content. |
| | Here are some demonstration: |
| | {Demonstrates} |
| | Here is the question and its corresponding page: |
| | Question:{Question} Page:{Page} |
| | — |
| | reply: |

| Slots | Question | Question about the given web page from WebSRC. |
|---|---|---|
| | Page | HTML souce code of the given web page from WebSRC. |
| | Demonstrates | The selected demonstration with its ground truth response. |

Table 7: The prompt for evaluating LLM on WebSRC.

| Prompt | You are a web scraper, you will extract the values of {Attributes} from a given web page segment. |
|---|---|
| | You are not allowed to use any tools, you can only use the information in the input JSON object. |
| | The web page is provided in the form of a JSON object, which contains several key-value pairs. Each key is a INDEX, and each value is another JSON object that has two fields: "XPATH" and TEXT. The "XPATH" field is a string that represents the location of an element in an HTML document. The "TEXT" field is a string that contains the text content of that element. |
| | The format of the answer, and do not reply other content: {Answer_Format}. |
| | Here are some demonstrations: |
| | {Demonstrates} |
| | Here is the given web page segment. |
| | JSON object: {Web_Page} |
| | — |
| | reply: |

| Slots | Attributes | Given attributes in the current vertical. |
|---|---|---|
| | Answer_Format | Response in the format based on the given attributes. |
| | Demonstrates | The selected demonstration with its ground truth response. |
| | Web_Page | The web page is provided in the form of a JSON object, which contains several key-value pairs. Each key is a INDEX, and each value is another JSON object that has two fields: "XPATH" and "TEXT". |

Table 8: The prompt for evaluating LLM on SWDE.

| | |
|---|---|
| Attributes | model", "price", "engine", "fuel_economy" |
| Web_Page | {1: {"XPATH": "/ html/ body/ div[2]/ div[3]/ div/ table/ tr/ td[2]/ div[1]/ div[1]/ div[1]/ div[1]/ h1/ br[1]", "TEXT": "2011 Kia Sportage"}, 2: {"XPATH": "/ html/ body/ div[2]/ div[3]/ div/ table/ tr/ td[2]/ div[1]/ div[1]/ div[1]/ div[2]/ div[1]/ div[2]/ div[1]/ table/ tr[1]/ td[2]/ b", "TEXT": "$18,295-24,795$"}, 3: {"XPATH": "/ html/ body/ div[2]/ div[3]/ div/ table/ tr/ td[2]/ div[1]/ div[1]/ div[1]/ div[2]/ div[1]/ div[2]/ div[1]/ table/ tr[2]/ td[2]", "TEXT": "$17,930-23,280$"}, , ... 36: {"XPATH": "/ html/ body/ div[2]/ div[3]/ div/ table/ tr/ td[2]/ div[1]/ div[4]/ div[6]/ div/ div[4]/ table/ tr[5]/ td[1]", "TEXT": "2011 Kia Sportage"}} |
| Answer_Format | model": [INDEX1, INDEX2, ..., INDEXn] or []; "price": [INDEX1, IN-DEX2, ..., INDEXn] or []; "engine": [INDEX1, INDEX2, ..., INDEXn] or []; "fuel_economy":[INDEX1, INDEX2, ..., INDEXn] or []. |
| Demonstrates | JSON object: {1: {"XPATH": "/ html/ body/ div[2]/ div[3]/ div/ table/ tr/ td[2]/ div[1]/ div[1]/ div[1]/ div[1]/ h1/ br[1]", "TEXT": "2010 Toyota Sequoia"}, ... 20: {"XPATH": "/ html/ body/ div[2]/ div[3]/ div/ table/ tr/ td[2]/ div[1]/ div[4]/ div[3]/ table/ tr[17]/ td[2]", "TEXT": "19"}} —

reply:
model":[1]; "price":[2, 9]; "engine":[13]; "fuel_economy":[19, 20] |

Table 9: The values of all slots in a web page sampled from "auto" vertical. The "Web_Page" and "Demonstrates" values are partially hidden due to space limitations.