# OpenReview forum: "GEM: Gestalt Enhanced Markup Language Model for Web Understanding via Render Tree"
_EMNLP/2023/Conference — EMNLP 2023 Main_

### Official Review · Reviewer_bTQ3 · 2023-08-04

**Soundness:** 4

**Excitement:**

4: Strong: This paper deepens the understanding of some phenomenon or lowers the barriers to an existing research direction.

**Paper Topic And Main Contributions:**

This paper proposed a novel way to train a model that can be used on structure data like HTML. Borrowing idea from psychology, they proposed two pre-training task specifically designed to improve downstream performance on HTML datasets. One pre-training task is Binary Textual Style Prediction, which predicts whether two HTML nodes have the same style or not, and another pre-training task is Proximate Note Prediction, which predicts whether two HTML nodes are nearby or not. Both tasks takes text embeddings of the corresponding HTML nodes as input. After pre-training, they fine-tune the model on two HTML-related downstream task: web QA and web information extraction. The results show that the Gestalt-inspired pre-training methods indeed helped the downstream performance.

**Reasons To Accept:**

* Proposed a unique method that can enhance LM's ability to understand HTML text. This is important now as a lot of people is trying to enhance LLM with the ability to browse the web (e.g. WebGPT).
* Ablation studies showed that the proposed method is crucial to improve the final performance.

**Reasons To Reject:**

* Performance gain is relatively small compared to MarkUp LM, especially for web-info extraction experiment.

**Reproducibility:**

3: Could reproduce the results with some difficulty. The settings of parameters are underspecified or subjectively determined; the training/evaluation data are not widely available.

**Reviewer Confidence:**

3: Pretty sure, but there's a chance I missed something. Although I have a good feel for this area in general, I did not carefully check the paper's details, e.g., the math, experimental design, or novelty.

---

> ### Author Rebuttal · Authors · 2023-08-29
>
> We are grateful for your careful review of our paper and your constructive feedback. We are glad that you recognized our main contribution and ablation studies. Regarding your concern, we hope that our explanation below can address that.
>
> > Performance gain is relatively small compared to MarkupLM, especially for web-info extraction experiment.
>
> We appreciate your concern about the relatively small performance gain in the SWDE dataset. In order to enhance experiment validity, the experiment results for each vertical are an average of 10 training set permutations, and the final results are averaged across all verticals, resulting in an average of 80 experimental results (at lines 360-363). Meanwhile, it is noteworthy that MarkupLM has already demonstrated exceptional performance in SWDE (with an F1 score of 95.57%), and further efforts can only lead to marginal improvements. Additionally, as shown in Table 2, GEM achieves a smaller standard deviation than MarkupLM, indicating that visual information enhances the stability of GEM. We hope this explanation addresses your concern, and we will provide additional narratives to address this problem in the revision.
>
> Finally, we would like to express our gratitude again for your review.

---

### Official Review · Reviewer_W8C9 · 2023-08-05

**Soundness:** 5

**Excitement:**

4: Strong: This paper deepens the understanding of some phenomenon or lowers the barriers to an existing research direction.

**Paper Topic And Main Contributions:**

The paper presents a novel approach to training a pre-trained language model for modern webpages that require understanding the complex interplay between HTML structure, text and visual appearance and layout. The idea is based on a psychological theory called Gestalt laws of perception, one of which is about how humans perceive similar objects to form a group, and another is about how we perceive closely situated objects to form a coherent group. The authors apply these Gestalt laws of similarity and proximity of objects to the pre-training task by designing appropriate contrastive pre-training objectives, in addition to employing the more commonly used masked language modeling objective based on the text.

The beauty of the approach lies in 1) the ability to leverage the intuitive laws in infusing visual appearance information into the pre-training task for better page understanding (in addition to structure and text considered previously), and in doing so, 2) also circumventing the challenge of requiring visual "screenshots" as input which has been a big hurdle for all prior attempts that aimed to integrate text, markup and visual information in a single model.

The paper is clear in its exposition, giving the readers a solid motivation and intuition behind the proposed approach, and also in clearly demonstrating its benefits via comparative evaluation, ablation and a qualitative example analysis. Particularly interesting is the fact that the model doesn't require HTML rendering for fine-tuning or inference, which can be immensely useful for a practitioner in Information Extraction, Question Answering and other relevant areas that depend on page understanding. The paper clearly demonstrates how this is enabled by the Gestalt laws that can capture visual information useful toward downstream tasks without explicit incorporation of HTML structure.

Having said that, the paper can be improved by clarifying a few aspects around webpage preprocessing, data collection, sampling approach for training, ablation on the IE task, and including a discussion on error patterns that remain.

**Reasons To Accept:**

A. The paper introduces a novel pre-training approach based on a psychological theory that integrates visual information, HTML structure and text (markup and tokens) in a unified manner that ultimately delivers better performance on downstream tasks.

B. The paper is very clear in its exposition, right from motivation, introduction, situating itself in the broader work on the topic, methods, results and reflection.

C. The authors plan to publish the code and the model, which should promote further work using the idea, thereby stimulating more research in this relatively nascent area.



**Reasons To Reject:**

Please note the following are some avenues where the paper clarity can be improved; they are not necessarily strong reasons to reject.

A. Webpage processing - the paper uses a sliding window to truncate long webpages into sub-pages, the details of which are missing. How is a page split? Sharing this is critical to understanding how a HTML "section" does not inadvertently land into separate sub-pages.

B. Data collection - How are the 100k pages in the pre-training corpora selected from Common Crawl? Are there any websites more preferred than others, or is it a random sample? How do you ensure diversity, availability of modern pages that render well?

C. Sampling - The sampling of node pairs for both the pre-training objectives (STSP, PNP) is an imbalanced data problem. How do you sample negatives? How do you sample "hard" negatives (pairs that are actually negative, but may have similar perceptual appearance and/or proximity)?

D. Ablation - while visual information can be useful for an extractive Q&A task like in the case of WebSRC dataset, it can be particularly useful to see how it fares in the Information Extraction task because the latter involves discerning right elements from incorrect ones based on limited context and visual information. Was this experiment performed? It would be interesting to learn how well the approach shines here.

E. Error analysis - Finally, while the pre-training task clearly helps deliver better performance on both the tasks considered, the paper can do better in throwing light on pending error patterns, by including an analysis and/or discussion of few examples where the approach fell short.

**Reproducibility:**

4: Could mostly reproduce the results, but there may be some variation because of sample variance or minor variations in their interpretation of the protocol or method.

**Reviewer Confidence:**

5: Positive that my evaluation is correct. I read the paper very carefully and I am very familiar with related work.

---

> ### Author Rebuttal · Authors · 2023-08-29
>
> Thank you for your valuable feedback and your time in reviewing our paper. We appreciate your positive comments on our method's novelty, exposition, motivation, and intuition. We are committed to publishing the code and the model to facilitate further research in this area. We have provided the following explanation in response to your insightful suggestion about clarification.
>
> > Ablation - while visual information can be useful for an extractive Q&A task like in the case of WebSRC dataset, it can be particularly useful to see how it fares in the Information Extraction task because the latter involves discerning right elements from incorrect ones based on limited context and visual information. Was this experiment performed? It would be interesting to learn how well the approach shines here.
>
> Thank you for your question regarding the ablation experiment on the SWDE dataset. We perform this experiment but cannot include it in the paper because of the space limit. The table below shows the experimental results, which achieve similar results as on web question answering. We will provide this experiment in the appendix of our revised version.
>
> | # | Markup | STSP | PNP | Precision  | Recall     | F1          |
> |---|--------|------|-----|------------|------------|-------------|
> | 1 | √      |      |     | 95.99±1.49 | 95.16±1.70 | 95.57±1.65  |
> | 2 | √      | √    |     | 96.58±1.10 | 95.25±1.49 | 95.64±1.39  |
> | 3 | √      |      | √   | 96.62±1.20 | 95.48±1.73 | 95.80±1.64  |
> | 4 | √      | √    | √   | 96.84±1.42 | 95.66±1.59 | 96.04±1.53  |
>
> > Webpage processing - the paper uses a sliding window to truncate long webpages into sub-pages, the details of which are missing. How is a page split? Sharing this is critical to understanding how a HTML "section" does not inadvertently land into separate sub-pages.
>
> Thank you for your valuable suggestions. We employ the sliding window algorithm proposed by [1], which attempts to prune the nodes in the tree while preserving the relationship between the nodes. We will supplement these details in Section 3.1 of the revision and provide an additional narrative to help readers reproduce the experiment outcomes.
>
> > Data collection - How are the 100k pages in the pre-training corpora selected from Common Crawl? Are there any websites more preferred than others, or is it a random sample? How do you ensure diversity, availability of modern pages that render well?
>
> Thank you for your insightful questions. We select the 100k pages from Common Crawl using a two-step process. First, we filter out pages that are non-renderable using headless chrome. Second, we randomly sample 100k pages from the remaining pool. This way, we avoid any bias towards certain websites or domains, and ensure a diverse and representative sample of modern web pages that render well. We will provide more details on this process in Section 3.1 of our revision to clarify it. Because of space limitations, more data collection details will be available in the appendix.
>
> > Sampling - The sampling of node pairs for both the pre-training objectives (STSP, PNP) is an imbalanced data problem. How do you sample negatives? How do you sample "hard" negatives (pairs that are actually negative, but may have similar perceptual appearance and/or proximity)?
>
> We appreciate your feedback on sampling negative samples in our pre-training objectives. We follow a two-step procedure for both STSP and PNP to obtain balanced samples. First, we traverse and label all node pairs in a given web page as positive or negative, and store them in separate pools. Second, we randomly sample 100 samples from each pool (totaling 200 samples) for pre-training. If either pool has less than 100 samples, we use oversampling to ensure data balance. We agree that sampling hard negatives could improve our performance, and we intend to explore this direction in our future work. We apologize for omitting these details in the paper due to space constraints, but we will include them in the appendix of our revised version.
>
> > Error analysis - Finally, while the pre-training task clearly helps deliver better performance on both the tasks considered, the paper can do better in throwing light on pending error patterns, by including an analysis and/or discussion of few examples where the approach fell short.
>
> We are grateful for your insightful suggestions. In our revised version, we will include more analysis to illustrate the limitations and challenges of our method in certain scenarios.
>
>
>
> Once again, we would like to thank you for your constructive feedback. Your comments and suggestions were very insightful and helped us improve the quality of our work.
>
> # Reference
>
> [1] Deng, X., Shiralkar, P., Lockard, C., Huang, B., & Sun, H. (2022). Dom-lm: Learning generalizable representations for html documents. arXiv preprint arXiv:2201.10608.

---

### Official Review · Reviewer_6rWs · 2023-08-12

**Soundness:** 3

**Excitement:**

3: Ambivalent: It has merits (e.g., it reports state-of-the-art results, the idea is nice), but there are key weaknesses (e.g., it describes incremental work), and it can significantly benefit from another round of revision. However, I won't object to accepting it if my co-reviewers champion it.

**Missing References:**

@misc{gur2023understanding,
      title={Understanding HTML with Large Language Models},
      author={Izzeddin Gur and Ofir Nachum and Yingjie Miao and Mustafa Safdari and Austin Huang and Aakanksha Chowdhery and Sharan Narang and Noah Fiedel and Aleksandra Faust},
      year={2023},
      eprint={2210.03945},
      archivePrefix={arXiv},
      primaryClass={cs.LG}
}

**Paper Topic And Main Contributions:**

The paper provides a new method to pre-train language models to better handle markup language-based text, such as the content on web pages (which includes HTML and the styles associated with the elements). Two new pre-training tasks are suggested which aim to capture the rendered visual aspects of the markup - these tasks are inspired by Gestalt Psychology theory which states that components containing similar content are co-located and look similar. The paper hypothesizes this way of pre-training improves the model's ability to understand web page content, and shows this by the improved performance on web Q&A and information extraction task.

**Questions For The Authors:**

* A claim in the Introduction states "Regardless of render conditions, humans are capable of rapidly comprehending the content of rendered web pages, which indicates that human visual perception is an effective way to consume web semantics." Could you please point to any relevant citations for this? I'd like to understand what aspects of the human visual perception system are radically different from a regular OCR equipped vision model (especially one as large as CLIP or PaLI where the network has enough capacity to learn high-level concepts).
* There seem to be errors in the explanation of the STSP pretraining objective and the example provided in the paper to explain it. Based on the definition, nodes B and C in Figure 3 should be different, but they are marked as same. Could you please explain why that is, considering the background color is different?
* I think it's possible to achieve better POS results on GPT 3.5 with a little bit of fine-tuning or few-shot prompting. It is really good at reading comprehension, and given a large enough context length it should be able to get the right subset. The fact that it is able to generate the right ground truth tells me some prompt changes or post-processing might have gotten equal results. Were you able to try any alternate prompts or few-shot strategies?
* Apologies if I'm misunderstanding this, but based on the example in Table 8, it seems like the xpath for the elements does not contain any style information (whether specified inline in the HTML or via separate CSS files). Does that mean whatever mechanism you use to extract xpath for each element of the HTML does not return styling information? If the input to your transformer layers is completely devoid of any style information, I'd imagine the model would have a harder time learning those distributions. So I'm wondering how much of the improvements you see is the results of pretraining MarkupLM for longer vs your pretraining objectives. Do you have a comparison between MarkupLM and your model where the length of pretraining is the same?

**Reasons To Accept:**

The idea of using Gestalt Psychology theory as pre-training tasks has novelty and promise. This theory provides a computationally lightweight, reasonably accurate baseline assumption to make about the semantic connections between components on the page - and hence a reasonable proxy for using actual visual information about the page.

**Reasons To Reject:**

While I like the novelty of using Gestalt theory for pretraining tasks, there are a few shortcomings of this paper:
* The paper lacks adequate details on the pretraining performance, for e.g. comparison of the loss or perplexity compared to MarkupLM or RoBERTa on which this model is based. Considering the pre-training tasks are the main contributions of this paper, this is a significant omission.
* The paper does not provide adequate explanation or examples of Gestalt principles being flouted in web pages and it's effect on web Q&A and information extraction. A wikipedia page has section and para sub-headings - which are grouped closer together when in collapsed state. While looking similar and being co-located in this state, they are not semantically related (one section could be about Etymology whereas another about References). Similarly for news headlines, sports scores, etc.
* The paper compares performance of their pre-trained + fine-tuned encoder-decoder model against a larger decoder-only model (GPT 3.5) in a zero-shot untuned manner. This seems like an uneven comparison.
* I felt the paper is limited in scope to an improvement to a specific prior model in this domain. To establish the viability of the pre-training objective, it needed to show results on some more contemporary models, such as smaller-sized decoder-only models with similar fine-tuning as was done.

**Reproducibility:**

4: Could mostly reproduce the results, but there may be some variation because of sample variance or minor variations in their interpretation of the protocol or method.

**Reviewer Confidence:**

4: Quite sure. I tried to check the important points carefully. It's unlikely, though conceivable, that I missed something that should affect my ratings.

---

> ### Author Rebuttal · Authors · 2023-08-29
>
> Thank you for dedicating your time to reviewing our paper and offering insightful feedback. We are grateful for your recognition of the novelty and potential of our idea. About the concerns you raised, we have carefully considered them and would like to respond as follows.
>
> # Response to Concerns regarding GPT 3.5
>
> > I think it's possible to achieve better POS results on GPT 3.5 with a little bit of fine-tuning or few-shot prompting. It is really good at reading comprehension ... some prompt changes or post-processing might have gotten equal results. Were you able to try any alternate prompts or few-shot strategies?
>
> Thank you for raising this question. We conduct experiments with various prompts for WebSRC and discover that the one presented in Appendix A.2 produces optimal outcomes. The task description of this prompt is derived from the official WebSRC repository [2], which ensures that GPT-3.5 comprehends the task requirements. We will provide an additional narrative in our revision.
>
> We also agree that in-context learning (ICL) based few-shot strategies can be essential for this study, while similar approaches show remarkable performance in some tasks. Following your suggestions, to evaluate its effectiveness for WebSRC, we conduct a supplementary experiment using the prompt format suggested by [1]. This format consists of a task description, selected demonstrations, and a test instance. For each question-page pair, we randomly select $n$ demonstrations from the same vertical to ensure semantic relevance. Due to the token limit imposed by the OpenAI API, we set $n$ to 3. To minimize the influence of randomness, we repeat the experiment 5 times and report the average results. The following table presents the results:
>
> | Model       | EM    | F1    | POS    |
> |-------------|-------|-------|--------|
> | GPT-3.5     | 66.80 | 72.01 | 66.24  |
> | GPT-3.5-ICL | 66.99 | 71.92 | 68.43  |
> | GEM         | 67.04 | 72.79 | 87.00  |
>
>
> "GPT-3.5-ICL" denotes the result using the ICL prompt, and "GPT-3.5" and "GEM" results are from Table 3 of our paper. As illustrated by the table, GPT-3.5-ICL achieves a similar result to GPT-3.5 in terms of EM and F1, indicating that GPT-3.5's QA capability has been fully activated. Regarding POS, ICL shows modest improvement, but GEM still outperforms GPT-3.5 significantly, which implies GEM's superiority in consuming HTML structural information. We will include these experimental results in our revision for a complete comparison.
>
> > The paper compares performance of their pre-trained + fine-tuned encoder-decoder model against a larger decoder-only model (GPT 3.5) in a zero-shot untuned manner. This seems like an uneven comparison.
>
> We appreciate your observation regarding the comparison between GEM and GPT-3.5. The purpose of Section 4.4 is to demonstrate that GEM can achieve comparable results on tasks that require consuming HTML structure, which is a key focus of GEM, rather than to claim that GEM outperforms GPT-3.5 in every aspect.
>
> We acknowledge that the comparison can be further enhanced by leveraging model fine-tuning with task-specific training data. Therefore, to align these differences, we intend to fine-tune GPT-3.5 with the same dataset as GEM utilizing the OpenAI API, and we will host this result in the revision. However, the preliminary experiment indicates that fine-tuning GPT-3.5 cannot significantly improve its performance since it is already trained on massive web-like data, such as CommonCrawl [1]. We will validate this hypothesis through experiments and report the results in the revision.
>
> > I felt the paper is limited in scope to an improvement to a specific prior model in this domain. To establish the viability of the pre-training objective, it needed to show results on some more contemporary models, such as smaller-sized decoder-only models with similar fine-tuning as was done.
>
> We appreciate your concern regarding the viability of the Gestalt tasks. As illustrated in Table 1 and 2 of our paper, the Gestalt tasks can enhance the performance of RoBERTa and MarkupLM, two different architecture encoder-only PLMs. Furthermore, we would like to highlight that GEM is the current state-of-the-art PLM in this domain. Therefore, our experiments provide strong evidence for the viability of the Gestalt tasks. Additionally, we recognize the distinction between designing pre-training tasks on web understanding for encoder-only and decoder-only models. We are in the process of adapting the Gestalt tasks to decoder-only architectures, and this is a promising direction for future research.
>
> # Response to Concerns Regarding Pre-training
>
> > Apologies if I'm misunderstanding this, but based on the example in Table 8, it seems like the xpath for the elements does not contain any style information ...  Does that mean whatever mechanism you use to extract xpath ... does not return styling information? ...  I'd imagine the model would have a harder time learning those distributions. ... Do you have a comparison between MarkupLM and your model where the length of pretraining is the same?
>
> Thank you for your thoughtful questions. The XPath expressions GEM uses are standard and do not incorporate any stylistic information from the web pages. We would like to clarify that GEM does not require any stylistic information or additional visual input, as it only takes HTML and XPath as input. Unlike prior costly OCR-based efforts, GEM learns the visual perception from two Gestalt pre-training tasks, which use CSS properties as their ground truth labels. Moreover, GEM utilizes the visual information of rendered web pages only during the pre-training and does not need any rendering during fine-tuning or inference.
>
> To compare GEM and MarkupLM fairly, we experiment with the same pre-training condition (including pre-training steps and dataset). In Table 1 and 2,  "MarkupLM\*" denotes MarkupLM further pre-trained on our corpora with its original objective for the same pre-training steps. The gap between GEM and MarkupLM\* demonstrates the effectiveness of Gestalt tasks. We will provide additional narrative in the paper to make the experiment results more noticeable.
>
>
> > The paper lacks adequate details on the pretraining performance, for e.g. comparison of the loss or perplexity compared to MarkupLM or RoBERTa on which this model is based. Considering the pre-training tasks are the main contributions of this paper, this is a significant omission.
>
> We appreciate your insightful suggestion. We concur that comparing loss and perplexity would provide a more thorough evaluation of the model's pre-training performance. Due to space constraints, we are unable to include these results in the original version. We will manage to incorporate various figures of loss curves and a comprehensive analysis of perplexity.
>
> # Response to Rest Concerns
>
> > The paper does not provide adequate explanation or examples of Gestalt principles being flouted in web pages and it's effect on web Q&A and information extraction. A wikipedia page has section and para sub-headings - which are grouped closer together when in collapsed state. While looking similar and being co-located in this state, they are not semantically related ... Similarly for news headlines, sports scores, etc.
>
> We appreciate your comment and acknowledge that some web pages in our corpora do not follow the Gestalt principles. However, we would like to clarify that the Gestalt task is not the only pre-training task for GEM. It is an essential complementary task that leverages non-text information (i.e., vision) to enhance the existing text-based tasks (e.g., MLM). The Gestalt task is an enhancement, not a replacement, of the text-based tasks. Moreover, as [3] has shown, web pages that violate the Gestalt principles tend to have low survival rates on the internet. Hence, the proportion of noisy data samples is relatively low. We will provide more evidence to support this claim in our revision.
>
> Regarding the section headings and paragraph sub-headings, we believe that they are not very semantically related, but they have analogous semantic functions as headings. Nevertheless, we concede that the Gestalt task may be less effective on the example you gave. Therefore, we are intrigued by the work of [4], who have developed methods to eliminate noisy blocks from web pages (such as advertisements). We will cite their papers in future work and further optimize the model performance. Finally, we would like to emphasize that our empirical results (Table 1, 2, 4) demonstrate the effectiveness of our method based on the current corpora. We will provide an additional narrative in the revision to address this problem.
>
>
> > A claim in the Introduction states "Regardless of render conditions, ... , which indicates that human visual perception is an effective way to consume web semantics." Could you please point to any relevant citations for this? I'd like to understand what aspects of the human visual perception system are radically different from a regular OCR equipped vision model ...
>
> We agree that we should provide additional references to support this argument. Two studies, [3] and [5], endorse our claim on human perception. We will cite these references clearly in our revision to enhance its rigor.
>
> Regarding your question about the radical difference between human perception and the OCR-equipped vision model, we would like to clarify that our objective is to enhance the language model rather than the vision model. Nevertheless, we can identify two differences. First, Gestalt psychology is a prominent cognitive theory that is derived from human experiences of perceiving semantic groups, which motivates us to enhance language models with visual perception without additional visual input. In contrast, most vision models, such as CLIP or PaLI, aim to learn statistical alignment between modalities from large-scale data but do not explicitly model semantic groups.  Second, the input of GEM is HTML. OCR-equipped vision models take images as input and emphasize the importance of the fixed layout of the input image. However, the dynamic nature of web page rendering can lead to significant appearance variations across devices and browsers. Therefore, GEM is more suitable for web understanding. We will provide more detailed explanations in our revision.
>
> > There seem to be errors in the explanation of the STSP pretraining objective and the example provided in the paper to explain it. Based on the definition, nodes B and C in Figure 3 should be different, but they are marked as same. Could you please explain why that is, considering the background color is different?
>
> Following your question, we will enhance Figure 3 and make it more comprehensible. We would like to clarify that the background colors of example nodes, e.g., Node B and C, are intended to highlight them, and actually, they both have a white background on the web page. As stated in Section 3.2.1, Node B and C have the same textual style because they share the same *font, color*, and *background-color* CSS values. However, we acknowledge that this may not be clear to the readers and may create confusion about the original background color of example nodes. We will enhance the clarity of the figure to ensure that it is unambiguous and not misleading.
>
> Again, we would like to express our gratitude for evaluating our paper carefully. Thank you for your time and effort in providing a comprehensive review.
>
>
> # Reference
> [1]	Brown, T., Mann, B., Ryder, N., Subbiah, M., Kaplan, J. D., Dhariwal, P., ... & Amodei, D. (2020). Language models are few-shot learners. Advances in neural information processing systems, 33, 1877-1901.
>
> [2]	https://github.com/X-LANCE/WebSRC-Baseline
>
> [3]	Xiang, P., Yang, X., & Shi, Y. (2007, July). Web page segmentation based on gestalt theory. In 2007 IEEE International Conference on Multimedia and Expo (pp. 2253-2256). IEEE.
>
> [4]	Uma, R., & Latha, B. (2019). Noise elimination from web pages for efficacious information retrieval. Cluster Computing, 22, 14583-14602.
>
> [5]	Xu, Z., & Miller, J. (2016). Identifying semantic blocks in Web pages using Gestalt laws of grouping. World Wide Web, 19, 957-978.

---

### Meta-Review · Area_Chair_yWNJ · 2023-09-19

**Recommendation:** 5

**Metareview:**

This paper introduces a novel method for building pre-trained language models that are capable of handling markup language-based text and understanding the complex interplay between HTML structure, text and visual appearance, and layout. The idea behind the new approach is grounded in Gestalt laws of perception, which describe how humans perceive similar objects to form a group and how closely situated objects are also perceived to form a coherent group. The paper hypothesizes that this way of pre-training improves the model's ability to understand web page content, and the experiments confirm that this helps improve its performance on downstream tasks.

All reviewers agree that the paper has **a number of strengths and merits**. Specifically:
1. *Novelty and theoretical as well as practical appeal of the proposed idea*: This strength is highlighted by all reviewers.
2. *Overall clarity of writing*: Reviewer W8C9 points out that the paper is clear, well organized and well written.
3. *Reproducibility* is noted by all reviewers, and Reviewer W8C9 lists the fact that the authors plan to publish the code and the model among their reasons to accept.
4. In addition, Reviewer bTQ3 points out that *ablation studies* suggest that the proposed method is crucial to the improvement of the final performance of the model.

At the same time, all reviewers also identified **some weaknesses** and further areas for improvement, including:
1. *Lack of meaningful comparison and further model's analysis*: Reviewer 6rWs points out that comparison of the loss or perplexity of the proposed model to MarkupLM or RoBERTa on which the model is based is missing. In addition, Reviewer 6rWs has further feedback on the comparison to GPT 3.5.
2. *Evaluation and analysis of the results*: Reviewer 6rWs points out that it would be desirable for the paper to show improvements on a broader scale and using some more contemporary models. Furthermore, Reviewer W8C9 proposes improvements in terms of ablation studies and error analysis. Finally, Reviewer bTQ3 considers the performance gain to be relatively small compared to MarkupLM.
3. *Further methodological explanations*: Reviewer 6rWs also suggests that the paper should provide more detailed explanation and / or examples of Gestalt principles.
4. *Implementation details*: Finally, while formulated not as clear weaknesses, Reviewer W8C9 makes suggestions for further improvement and clarification of certain implementation details – see their review.

The authors did a good job addressing the concerns expressed and the questions posed by the reviewers. Most suggestions made by the reviewers have been acknowledged by the authors and seem to be likely to be integrated in the revised version of the paper.

---

### Decision · Program_Chairs · 2023-10-07

**Decision:**

Accept-Main

**Comment:**

This paper introduces a novel method for building pre-trained language models that are capable of handling markup language-based text and understanding the complex interplay between HTML structure, text and visual appearance, and layout. The idea behind the new approach is grounded in Gestalt laws of perception, which describe how humans perceive similar objects to form a group and how closely situated objects are also perceived to form a coherent group. The paper hypothesizes that this way of pre-training improves the model's ability to understand web page content, and the experiments confirm that this helps improve its performance on downstream tasks.

All reviewers agree that the paper has **a number of strengths and merits**. Specifically:
1. *Novelty and theoretical as well as practical appeal of the proposed idea*: This strength is highlighted by all reviewers.
2. *Overall clarity of writing*: Reviewer W8C9 points out that the paper is clear, well organized and well written.
3. *Reproducibility* is noted by all reviewers, and Reviewer W8C9 lists the fact that the authors plan to publish the code and the model among their reasons to accept.
4. In addition, Reviewer bTQ3 points out that *ablation studies* suggest that the proposed method is crucial to the improvement of the final performance of the model.

At the same time, all reviewers also identified **some weaknesses** and further areas for improvement, including:
1. *Lack of meaningful comparison and further model's analysis*: Reviewer 6rWs points out that comparison of the loss or perplexity of the proposed model to MarkupLM or RoBERTa on which the model is based is missing. In addition, Reviewer 6rWs has further feedback on the comparison to GPT 3.5.
2. *Evaluation and analysis of the results*: Reviewer 6rWs points out that it would be desirable for the paper to show improvements on a broader scale and using some more contemporary models. Furthermore, Reviewer W8C9 proposes improvements in terms of ablation studies and error analysis. Finally, Reviewer bTQ3 considers the performance gain to be relatively small compared to MarkupLM.
3. *Further methodological explanations*: Reviewer 6rWs also suggests that the paper should provide more detailed explanation and / or examples of Gestalt principles.
4. *Implementation details*: Finally, while formulated not as clear weaknesses, Reviewer W8C9 makes suggestions for further improvement and clarification of certain implementation details – see their review.

The authors did a good job addressing the concerns expressed and the questions posed by the reviewers. Most suggestions made by the reviewers have been acknowledged by the authors and seem to be likely to be integrated in the revised version of the paper.